# Effects of pre-transport diet, transport duration and transport condition on immune cell subsets, haptoglobin, cortisol and bilirubin in young veal calves

Francesca Marcato[1,2]*, Henry van den Brand[1], Christine A. Jansen[3], Victor P. M. G. Rutten[3,4], Bas Kemp[1], Bas Engel[5], Maaike Wolthuis-Fillerup[2], Kees van Reenen[2]

1 Adaptation Physiology Group, Wageningen University & Research, Wageningen, The Netherlands,
2 Wageningen Livestock Research, Wageningen University & Research, Wageningen, The Netherlands,
3 Faculty of Veterinary Medicine, Department of Infectious Diseases and Immunology, Utrecht University, Utrecht, The Netherlands, 4 Faculty of Veterinary Science, Department of Veterinary Tropical Diseases, University of Pretoria, Republic of South Africa, 5 Biometris, Wageningen University & Research, Wageningen, The Netherlands

* francesca.marcato@wur.nl

**Data Availability Statement:** All relevant data are within the manuscript and its Supporting Information files.

## Abstract

The aim of this study was to investigate effects of pre-transport diets, transport durations and transport conditions on immune cell subsets, haptoglobin, cortisol and bilirubin of young calves upon arrival at the veal farm. An experiment was conducted with a 2 × 2 × 2 factorial arrangement with 3 factors: 1) provision of rearing milk or electrolytes at the collection center (CC); 2) transport duration (6 or 18 hours) and 3) transport condition (open truck or conditioned truck). Holstein-Friesian and cross-bred calves were used (N = 368; 18 ± 4 days; 45.3 ± 3.3 kg). Blood samples were collected from calves (N = 128) at the collection center, immediately post-transport (T0) and 4, 24, 48 hours, week 1, 3 and 5 post-transport. Blood was analyzed for cortisol, bilirubin, haptoglobin, IgG and IgM. Moreover, cell counts of neutrophils, lymphocytes, monocytes, basophils and eosinophils were measured in blood samples taken at the collection center and T0. In these same blood samples, different lymphocyte populations were characterized by flow cytometry, including CD14+ cells, NK cells, δγ+ T cells, CD8+ cells, CD4+ cells and CD21+ cells. Calves transported in the conditioned truck had higher amounts of white blood cell count (WBC) (Δ = 1.39 × 10⁹/l; P = 0.01), monocytes (Δ = 0.21 × 10⁹/l; P = 0.04), neutrophils (Δ = 0.93 × 10⁹/l; P = 0.003), than calves transported in the open truck regardless, of pre-transport diet or transport duration. The study showed that transport condition and duration influenced parts of the innate immune system of young veal calves. Cortisol, bilirubin and WBC seemed to be connected by similar underlying mechanisms in relation to transport conditions. However, it is unclear which specific pathways in the immune system of young calves are affected by different transport conditions (e.g. temperature, humidity, draught).

**Funding:** This study was financially supported by Stichting Brancheorganisatie Kalversector (SBK) (www.kalversector.nl) and the Dutch Ministry of Agriculture, Nature and Food Quality (www.minlnv. nl). The funders had no role in study design, data collection and analysis, decision to publish, or preparation of the manuscript.

**Competing interests:** The authors have declared that no competing interests exist.

## Introduction

Transport represents a challenge for calves [1]. Veal calves are transported when they are 14 to 20 days old, and particularly at this young age, they are vulnerable to transport stress [2,3]. In fact, the immune system of young calves is not fully developed and calves lack the time necessary for building immunocompetence compared to older cattle [2,4]. During handling, loading, and commingling around transport, calves may be exposed to microorganisms against which they have no colostral antibodies [5]. Because transport represents a severe stressor for calves, it may impede immunocompetence and enhance the susceptibility of calves to diseases [6]. Additionally, stress may result in a disruption of the balance between humoral and cellular components of the immune system [7]. To our knowledge, there are limited and contrasting results on which part of the immune system of young calves is affected by transport. Following transport, for example, percentages of γδ T cells, that represent up to 60% of circulating T-cells in young calves, have been reported to increase [8,9]. However, Riondato et al. [10] showed a decrease in percentages of these cell populations, whereas Baldwin et al. [11] reported a significant fluctuation in their proportion. Transport related stress may also induce changes in antibody responses. Mackenzie et al. [12] showed an increase in total $IgG_1$ and $IgG_2$ levels following transport and weaning of calves. In contrast, Mormede et al. [13] found no acute effects of transport on immunoglobulins levels. It is also unknown how specific immune cell subsets of young calves respond in terms of functionality to transport. Moreover, information on the influence of several transport-related factors, namely pre-transport diet, transport duration, and transport conditions on immune cells of young calves is limited. The most common changes observed after transport of all age groups of cattle include an increase in total white blood cell numbers (WBC), basophils, and neutrophils and a decrease in lymphocytes, eosinophils and monocytes [4]. These changes were mainly investigated in relationship to transport duration, whereas other factors associated with transport, such as pre-transport diet or transport conditions were never studied. Other literature studies showed that also acute phase proteins (APPs) seemed to be influenced by transport duration. Serum concentrations of haptoglobin and serum amyloid-A (SAA) and fibrinogen of cattle were enhanced following transport [14,15]. After long transport duration (> 24 hours), Murata and Miyamoto [16] and Arthington et al. [17] showed increased serum haptoglobin concentrations in calves. In contrast, Buckham Sporer et al. [18] observed decreased concentrations of haptoglobin and fibrinogen after 9 hours transport of bulls (282 ± 4 days of age). The majority of this type of work is done on adult animals or at least in older animals. It can, however, be expected that younger calves are more vulnerable to transport-related challenges. This might be expressed in changes in immune related variables, such as immune cell subsets and APPs. The aims of the current study were to investigate immune responses (expressed by changes in immune cell subsets, haptoglobin) and changes in bilirubin and cortisol concentrations in blood of young calves in relation to three transport-related factors (pre-transport diet, transport duration and transport condition).

## Materials and methods

### Animals and experimental design

The experiment was approved by the Central Committee on Animal Experiments (the Hague, the Netherlands; approval number 2017.D-0029). In total, 368 male Holstein-Friesian and crossbred calves (18 ± 4 days; 45.3 ± 3.3 kg body weight (BW)), divided over two consecutive batches (N = 184/batch) were used. The experiment had a 2 × 2 × 2 factorial arrangement with 3 factors: 1) provision of rearing milk or electrolytes at the collection center (CC); 2) transport duration (6 or 18 hours), and 3) transport condition (open truck or conditioned truck).

In this study we used calves as they normally exist in practice, thus they were collected from different dairy farms in Germany, transported first to a collection center located in Bocholt-Barlo, Germany, and then towards a veal farm in Veghel, The Netherlands. All calves were complying with the minimal weight and health status requirements (BW > 36 kg; age: minimum 14 days; no signs of disease and injury) [19]. Since we used animals which followed common procedures of collection, mixing and transport, calves most likely were already challenged prior to their arrival at the collection center (e.g. they had been subjected to feed and water withdrawal, and various handling and transport procedures). Therefore, pre-transport blood values shown in this paper most likely are not representative of baseline values of calves.

## Procedures at the collection center, transport and at the veal farm

At the collection center, calves were randomly allocated by the manager of the collection center to one of eight treatment groups (N = 23 calves per treatment group per batch). Within each batch, 8 calves per treatment group were randomly selected for blood sampling (performed by personnel that were blinded to the treatment groups). After blood sampling, calves were fed via a bucket with nipples with either 1.5 l of rearing milk (125 g of milk powder/l; ME = 4028 kcal/kg of milk powder, CP = 190 g/kg, digestible lysine = 18.7 g/kg; Tentofok KO, Tentego, The Netherlands) or a mixture of electrolytes (20 g of electrolytes/l of water; Navobi, Staverden, The Netherlands) dissolved in 1.5 l water. No refusals were recorded, thus all calves ingested the expected amount of milk or electrolytes.

After feeding and before transport, calves had the opportunity to rest for 2 hours and after that they were loaded on the vehicle. The vehicle consisted of two parts, a truck and a trailer. The truck was conditioned, which means it was provided with a side-ventilation system, it was insulated and the climate was controlled with regard to in and outlet of air (KVM Livestock Transport System$^{TM}$, Kleventa BV, Lichtenvoorde, The Netherlands). Settings were according to those provided by the manufacturer and applied by the transporter. The trailer was regular, open, and lacked a ventilation system or climate control. Both the truck and trailer were divided into 4 compartments with straw bedding, two at the lower deck (3.60 m length × 2.45 m width × 1.35 m height) and two at the upper deck (3.60 m length × 2.45 m width × 1.45 m height). Additionally, compartments were arranged in both front and back part of the truck and trailer. Each compartment contained 23 calves and had the same stocking density (0.383 m$^2$ per calf). Each compartment contained calves of one of the eight treatments. Treatments were divided over the truck and the trailer in a way that all main treatments were positioned at the different parts of the truck and trailer (see S1 Fig). The actual temperature and relative humidity inside the truck and trailer were recorded by loggers (N = 8; Escort imini, Cryopak Verification Technologies, Inc.). Each logger was positioned in the middle of each compartment of the truck and trailer and recorded temperature and humidity every 10 minutes. The average (and range in) temperature and relative humidity during transport in the conditioned and open truck are shown in Table 1. After loading of calves in the vehicle, transport started

**Table 1. Mean and range (between brackets) of actual temperature (T) and relative humidity (RH) inside the conditioned and open trucks during short (6 hours) or long (18 hours) transport of young calves to the veal farm.**

| | Conditioned truck | | | | Open truck | | | |
|---|---|---|---|---|---|---|---|---|
| | Batch 1 | | Batch 2 | | Batch 1 | | Batch 2 | |
| | T (˚C) | RH (%) | T (˚C) | RH (%) | T (˚C) | RH (%) | T (˚C) | RH (%) |
| 6 h | 9.2 (8.2–10.3) | 66.0 (61.0–75.1) | 13.0 (11.7–13.9) | 74.1 (65.0–81.0) | 7.4 (6.2–9.1) | 74.1 (66.3–84.2) | 11.5 (10.4–12.5) | 80.3 (67.6–88.8) |
| 18 h | 7.8 (4.5–11.2) | 68.2 (58.5–78.9) | 13.6 (11.2–16.3) | 77.9 (65.4–83.9) | 6.6 (3.9–9.6) | 75.8 (66.3–86.5) | 14.0 (10.8–16.6) | 77.3 (66.2–86.2) |

with two drivers. Drivers switched every 3 hours. No food or water were provided to calves during transport. After 6 hours transport, the truck arrived at the veal farm and all calves were unloaded. Calves assigned to 6 hours transport were placed in the veal farm, whereas the calves assigned to 18 hours transport were reloaded on the truck and trailer (in the same compartments as before) and transported for another 12 hours. At the veal farm, a total of 64 pens were available, divided over 8 similar compartments, each containing 8 pens, with 5 or 6 calves per pen. Within each compartment, treatments were randomly distributed across pens. Each pen contained 2 calves that were already used for blood sampling at the collection center, and these calves were sampled again immediately after placement at the veal farm (T0). After blood collection at T0, all calves received electrolytes (20 g of electrolytes/l of water; Navobi, Staverden, The Netherlands) dissolved in 3 l of water. For the first 3 weeks, calves were housed individually in temporary partitions positioned within each pen, but these partitions were open, thus calves were able to interact to certain extents with each other. After 3 weeks, the partitions were removed and calves were kept in groups. All calves included in the current research followed the normal production cycle of the veal industry, at the end of which calves were slaughtered and their meat was used for human consumption.

More details on the time and nutrient composition of first feeding at the veal farm are provided by Marcato et al. [20].

## Blood collection and analysis

To determine effects of transport on subsets of blood immune cells, samples were collected at different sampling moments: at the collection center, prior to feeding (CC), upon arrival at the veal farm prior to provision of electrolytes (T0), after 4, 24, 48 hours (T4, T24, T48) and at week 1, 3 and 5 post-transport. Blood samples (10 ml) were collected from the jugular vein into different vacutainer tubes (Vacuette, Greiner BioOne, Kremsmunster, Austria) and kept at room temperature before centrifugation (3,000 × g for 15 min at 4˚C). Plasma and serum were then decanted and stored at– 20˚C until analysis. Fluorescence flow cytometry (XT1800VET, IDEXX Bioresearch) was used to determine absolute numbers of different cell types in full blood, including WBC, neutrophils, lymphocytes, monocytes, eosinophils and basophils. Plasma samples were analyzed for levels of immunoglobulins, bilirubin, and haptoglobin, and all measurements were done in duplicate. Immunoglobulin IgG and IgM were measured by an indirect enzyme-linked immunosorbent assay (ELISA) against keyhole limpet hemocyanin (KLH). Plates were coated with 4 μg/ml of KLH (100 μl/well). Natural antibodies of the IgG isotype were detected in plasma, using 1: 20,000 diluted sheep polyclonal anti-bovine IgG-heavy chain conjugated to horseradish PO (Cat. No. E10-118P, Bethyl Laboratories). Natural antibodies of the IgM isotype were detected in diluted plasma (1:40 as starting dilution) using 1: 20,000 diluted rabbit polyclonal anti-bovine IgM conjugated to horseradish PO (Cat. No. A10-100P, Bethyl Laboratories). Titers were calculated as log2 values of the dilutions, in accordance to Mayasari et al. (2015). Plasma bilirubin concentrations were measured at the mass spectrophotometer, using kit n. 10012, Bilirubin liquicolor, from Human (Wiesbaden, Germany). Plasma haptoglobin concentration was determined with kit n. TP801 from Tridelta Development Ltd. (Maynooth, Ireland). Radioimmunoassay (RIA) (kit n. IM1841, Beckman Coulter, Czech Republic) was used to detect cortisol in plasma samples.

A heparin tube (5 ml) was used for isolation of mononuclear white blood cells. Blood (2 ml) was diluted in phosphate buffered saline (PBS) (2 ml). Diluted blood (2 ml) was then gently added on the top of 2 ml Histopaque (1.083) and samples were centrifuged (1,000 × g, 45 min). White blood cells were then collected from the interphase, resuspended in 2 ml PBS and centrifuged for 20 s at maximum speed (10,000 × g) in the Eppendorf centrifuge. The washing

step was repeated twice and the pellet was then suspended in 1 ml fetal calf serum (FCS). An equal volume of freezing medium (1 ml, 10% FCS-RPMI 1640 + 20% DMSO) was added drop by drop to cells. Samples were then stored immediately in a freezing container (Mr. Frosty, Nalgene®) at -80°C.

## Flow cytometry

Different combinations of monoclonal antibodies (mAbs) were used to characterize lymphocyte subsets, using flow cytometry (Table 2). Mix 1 included mouse anti-bovine CD335-AF488 (AKS1, Bio-Rad, diluted 1:10) and mouse anti-bovine CD8-PE (CC63, Bio-Rad, diluted 1:10). Mix 2 included mouse anti-bovine WC1-FITC (CC15, Bio-Rad, diluted 1:50) and mouse anti-bovine CD4-PE (CC8, Bio-Rad, diluted 1:10). Mix 3 included mouse anti-bovine CD14-FITC (CC-G33, Bio-Rad, diluted 1:50), mouse anti-bovine CD21-PE (CC21, Bio-Rad, diluted 1:10), mouse anti-bovine MHC class II (CC302, Bio-Rad, diluted 1:50) and goat-anti-mouse-IgG2a-APC (diluted 1:2,000).

For the staining of cell surface markers, white blood cells were thawed, resuspended in 15 ml medium (10% FCS-RPMI 1640) and centrifuged (1300 × rpm, 5 min, 4°C). Supernatant was discarded and the washing step was repeated. Cells were suspended in 1 ml medium and counted. A total of approximately 500,000 cells/well was transferred to 96 well round bottom plate and PBA (PBS supplemented with 0.5% BSA and 0.005% NaAz) was added. The plate was centrifuged (1,300 × rpm, 2 min, 4°C), and supernatant was discarded. Each mix of antibodies (50 μl/well) was added to the plate and cells were incubated in the dark for 20 min, on ice. The plate was then washed with PBA and centrifuged (1,300 × rpm, 2 min, 4°C). At this stage, cells that were stained with mix 1 and 2 were permeabilized with a permeabilization mix (1 volume FACS permeabilizing solution (BD Pharmingen) + 1 volume FACS lysing solution (BD) + 8 volumes milliQ water). Cells stained with mix 3 remained on ice. To investigate whether the function of CD8+, CD4+ T cells, NK cells and γδ T cells was affected by transport-related stress, intracellular expression of perforin was determined. Therefore, after permeabilization, 50 μl prediluted (1:10) mouse anti-human perforin AF647 (δG9, BD Pharmingen) was added to cells stained with mix 1 and 2. Cells, that had previously received mix 3, received 50 μl prediluted (1:2,000) goat-anti-mouse-IgG2a-APC. Cells were incubated in the dark for 20 min, on ice. Cells were washed with PBA and centrifuged (1,300 × rpm, 2 min, 4°C) and then resuspended in 200 μl PBA/well.

**Table 2. Composition of the different mixes of monoclonal antibodies used for the flow cytometry analysis.**

|  | Antibody name | Target |
|---|---|---|
| **Mix 1** | Mouse anti bovine CD8: RPE | CD8+ T cells |
|  | Mouse anti bovine CD335: AF488 | NK cells |
|  | Mouse anti human perforin AF647[1] | Perforin[1] |
| **Mix 2** | Mouse anti bovine WC1: FITC | gamma delta T cells |
|  | Mouse anti bovine CD4:PE | CD4+ T cells |
|  | Mouse anti human perforin AF647 | Perforin[1] |
| **Mix 3** | Mouse anti bovine CD14: FITC | monocytes |
|  | Mouse anti bovine CD21:PE | B cells |
|  | Mouse anti bovine MHC Class II UNL | - |
|  | Goat-anti-mouse-IgG2a-APC | IgG2a |

[1]Perforin = it indicates the degranulation of NK cells or T cells, thus is a marker for activation of these cells.

A minimum of 200,000 cells from a gated lymphocyte population based on FSC and SSC scatter was acquired and measured on a FACS flow cytometer (BD FACSCanto II). Data were then analyzed on FlowJo v10 to obtain percentages of different lymphocyte subsets.

## Statistical analyses

All statistical analyses were performed with SAS 9.4 (SAS Inst. Inc., Cary, NC). First, the analyses of data on immediate post-transport (T0), including WBC, neutrophils, lymphocytes, monocytes, eosinophils, basophils, bilirubin, haptoglobin, cortisol, and FACS data (mix 1, 2, 3) are explained. Continuous data, such as bilirubin, were analyzed with a linear mixed model (analysis with restricted maximum likelihood with SAS procedure PROC MIXED). Residuals were checked for normality and homogeneity of variance and variables were log-transformed when needed. Data expressed as proportions, such as FACS data, were analyzed with a generalized linear mixed model (analysis with penalized quasi likelihood with SAS procedure GLIMMIX), with a logit link function, specifying the "error" variance as a multiple of the binomial variance. Both the linear mixed model and the generalized linear mixed model comprised the following fixed effects in the systematic part of the model (the linear predictor part):

$$Y = \mu + \text{Batch}_i + \text{Uplo}_j + \text{Bafr}_k + \text{Diet}_l + \text{Type}_m + \text{Duration}_n + (\text{Diet}_l \times \text{Duration}_n)$$
$$+ (\text{Diet}_l \times \text{Type}_m) + (\text{Duration}_n \times \text{Type}_m) + (\text{Diet}_l \times \text{Type}_m \times \text{Duration}_n) \quad (1)$$

Where: Y = dependent variable, $\mu$ is the overall mean, and $\text{Batch}_i$ = batch (i = 1, 2), $\text{Uplo}_j$ = position in the vehicle (j = upper or lower deck), $\text{Bafr}_k$ = position in the vehicle (k = front or back), $\text{Diet}_l$ = diet at the collection center (l = rearing milk or electrolytes), $\text{Type}_m$ = Transport condition (m = open or conditioned truck), and $\text{Duration}_n$ = transport duration (n = 6 or 18 hours). The model also comprised two- and three-way interactions between diet, transport condition and transport duration. Interactions were considered not significant when $P > 0.05$. In addition, random effects for pen and compartment at the veal farm were included (in the linear predictor). Here and in the subsequent analyses, for all fixed effects, approximate F-tests were used [21]. Interactions that were not significant were excluded from the model (when higher order interactions were already excluded, i.e. respecting the hierarchy of interaction terms) and subsequent pairwise comparisons were done with Fisher's LSD method.

Second, data on bilirubin, haptoglobin and cortisol from T0 until week 3 were analyzed with a linear mixed model for continuous data. The systematic part of this model comprised the following fixed effects:

$$Y = \mu + \text{Batch}_i + \text{Uplo}_j + \text{Bafr}_k + \text{Diet}_l + \text{Type}_m + \text{Duration}_n + \text{Time}_o + (\text{Diet}_l$$
$$\times \text{Duration}_n) + (\text{Diet}_l \times \text{Type}_m) + (\text{Duration}_n \times \text{Type}_m) + (\text{Diet}_l \times \text{Time}_o)$$
$$+ (\text{Duration}_n \times \text{Time}_o) + (\text{Type}_m \times \text{Time}_o) + (\text{Diet}_l \times \text{Type}_m \times \text{Duration}_n) \quad (2)$$

Where: Y = dependent variable, $\mu$ is the overall mean, and $\text{Batch}_i$ = batch (i = 1, 2), $\text{Uplo}_j$ = position in the vehicle (j = upper or lower deck), $\text{Bafr}_k$ = position in the vehicle (k = front or back), $\text{Diet}_l$ = diet at the collection center (l = rearing milk or electrolytes), $\text{Type}_m$ = transport condition (m = open or conditioned truck), $\text{Duration}_n$ = transport duration (n = 6 or 18 hours), and $\text{Time}_o$ = sampling moment (o = CC, T0, T4, T24, T48, week 1 and 3). Three-way interactions between diet, transport condition, transport duration, and two-way interactions between diet, transport condition, transport duration and time were also included in the model. Interactions were considered not significant when $P > 0.05$. The model comprised random pen, compartment, and animal effects. For the animal effects a first order auto regressive

model (based on the actual distance between time points) was adopted to introduce correlation in the model between repeated measurements on the same animal.

Third, data on bilirubin, haptoglobin and cortisol of week 5 were analyzed, using model 2. Although calves were housed individually in baby boxes until week 3, they were in group pens from week 5 onward, so random pen effects were included in the model.

Finally, differences between pre- and post-transport measurements (deltas, Δ = T0 –CC) were calculated for WBC, neutrophils, lymphocytes, monocytes, eosinophils, basophils, bilirubin, haptoglobin, cortisol, and FACS variables. These differences were also analyzed, using model 2.

Although no treatments were applied yet at the collection center, a preliminary statistical test, using model 1 but without interactions, was performed to investigate whether groups of calves already differed before treatments started. No statistical differences were found between treatment groups.

## Results

Three-way interactions and two-way interactions between pre-transport diet and transport condition and between transport duration and transport condition were never significant. Consequently, results will be shown as follows: 1) Two-way interaction between pre-transport diet and transport duration (Table 3); 2) Main effects of pre-transport diet, transport condition and transport duration (Tables 4–6); 3) Effects of treatments in time until 5 week post-transport (Figs 1, 2, 3A and 3B). Within the first two parts, results will be presented at 2 levels: 1) Effects of treatments immediately post-transport (T0) (Tables 3–5); 2) Effects of treatments on differences between pre and post-transport measurements (deltas, Δ = T0 –CC; Tables 3 and 6).

### Interaction between pre-transport diet and transport duration

**Effects on variables immediately post-transport (T0).** An interaction between pre-transport diet and transport duration was detected for bilirubin ($P$ = 0.03; Table 3). Both milk-fed and electrolyte-fed calves transported for 18 hours and electrolyte-fed calves transported for 6 hours had similar higher bilirubin concentrations than milk-fed calves transported for 6 hours (Δ = 4.24 μmol/l on average) directly post-transport.

**Effects on the difference between pre- and post-transport measurements (Δ = T0 – CC).** An interaction between pre-transport diet and transport duration was found for deltas

**Table 3. Interaction between pre-transport diet (milk vs electrolytes) and transport duration (6 vs 18 hours) detected in blood parameters measured immediately post-transport (T0) (A) and on the difference between pre- (CC) and post-transport (T0) measurements (B, deltas, Δ = T0 –CC) in young veal calves (LS means).**

| A. Immediate post-transport (T0) | | | | | |
|---|---|---|---|---|---|
| Parameter | Milk | | Electrolytes | | SEM[1] | $P$-value Interaction |
| | 6 h | 18 h | 6 h | 18 h | | |
| Bilirubin, μmol/l | 8.39[b] | 12.35[a] | 13.24[a] | 12.30[a] | 1.21 | 0.03 |
| B. Δ = T0 –CC | | | | | |
| Parameter | Milk | | Electrolytes | | SEM | $P$-value Interaction |
| | 6 h | 18 h | 6 h | 18 h | | |
| Basophils, 10⁹/l | 0.22[a] | 0.15[ab] | 0.19[b] | 0.16[ab] | 0.02 | 0.03 |
| Bilirubin, μmol/l | -3.75[b] | 1.83[a] | 3.64[a] | 1.50[a] | 1.35 | 0.02 |
| Haptoglobin, mg/ml | -0.04[ab] | 0.11[a] | 0.03[ab] | -0.05[b] | 0.05 | 0.05 |

[a-b] Least square means within a row lacking a common superscript differ ($P \leq 0.05$).

[1]SEM = pooled standard error.

of basophils, bilirubin and haptoglobin (Table 3). Milk and electrolyte-fed calves transported for 18 hours increased to the same degree their number of basophils and bilirubin concentration. However, after 6 hours transport, milk-fed calves had a stronger increase in the number of basophils than electrolyte-fed calves ($P = 0.03$). In addition, milk-fed calves had a decrease in bilirubin concentration compared with electrolyte-fed calves ($P = 0.02$). Differences in haptoglobin concentrations after 6 hours transport were similar between milk-fed calves and electrolyte-fed calves. However, milk-fed calves and transported for 18 hours increased their haptoglobin values, whereas electrolyte-fed calves and transported for 18 hours had a decrease ($P = 0.05$).

## Main effects of pre-transport diet, transport condition, and transport duration

**Effects on variables immediately post-transport (T0).** As shown in Table 4, diet had an effect on concentrations of monocytes and cortisol at T0. Milk-fed calves had higher monocytes concentration ($\Delta = 0.21 \times 10^9$/l) and lower cortisol than electrolyte-fed calves ($\Delta = -2$ µmol/l). Moreover, milk-fed calves had a lower proportion of CD8+ lymphocytes than electrolyte-fed calves ($\Delta = -1.19\%$, Table 5).

Transport condition influenced absolute numbers of WBC, monocytes, neutrophils, eosinophils and basophils at T0, and also proportion of neutrophils and lymphocytes relative to WBC (Table 4). With regards to absolute levels, calves transported in the conditioned truck had higher amount of WBC ($\Delta = 1.39 \times 10^9$/l), and higher neutrophils ($\Delta = 0.93 \times 10^9$/l),

**Table 4. Effects of diet composition (electrolytes vs milk) at the collection center, transport conditions (conditioned vs open truck) and transport duration (6 vs 18 hours) on immune parameters of young calves measured directly post-transport (LS means).**

| Parameter | Pre-transport diet | | | | Transport condition | | | | Transport duration | | | |
|---|---|---|---|---|---|---|---|---|---|---|---|---|
| | Electrolytes | Milk | SEM[1] | *P*-value | Conditioned | Open | SEM | *P*-Value | 6 h | 18 h | SEM | *P*-Value |
| WBC[2], $10^9$/l | 8.19 | 9.44 | 0.39 | 0.11 | 9.51 | 8.12 | 0.40 | 0.01 | 8.54 | 9.10 | 0.41 | 0.79 |
| Monocytes, $10^9$/l | 1.18 | 1.39 | 0.07 | 0.03 | 1.39 | 1.18 | 0.07 | 0.04 | 1.27 | 1.30 | 0.09 | 0.87 |
| Monocytes, %[3] | 14.16 | 15.13 | 0.51 | 0.16 | 14.60 | 14.69 | 0.51 | 0.89 | 14.72 | 14.56 | 0.63 | 0.86 |
| Lymphocytes, $10^9$/l | 3.03 | 3.27 | 0.16 | 0.18 | 3.15 | 3.15 | 0.16 | 0.99 | 3.22 | 3.08 | 0.18 | 0.59 |
| Lymphocytes, % | 38.22 | 37.40 | 1.36 | 0.77 | 35.44 | 40.17 | 1.36 | 0.03 | 38.45 | 37.15 | 1.68 | 0.57 |
| Neutrophils, $10^9$/l | 3.34 | 3.94 | 0.25 | 0.25 | 4.10 | 3.17 | 0.25 | 0.003 | 3.53 | 3.75 | 0.31 | 0.79 |
| Neutrophils, % | 39.60 | 39.55 | 1.29 | 0.88 | 41.48 | 37.64 | 1.26 | 0.02 | 37.92 | 41.37 | 1.27 | 0.40 |
| Eosinophils, $10^9$/l | 0.49 | 0.65 | 0.09 | 0.50 | 0.69 | 0.45 | 0.09 | 0.13 | 0.55 | 0.60 | 0.09 | 0.54 |
| Eosinophils, % | 5.68 | 5.83 | 0.59 | 0.86 | 6.23 | 5.27 | 0.59 | 0.46 | 5.49 | 6.01 | 0.73 | 0.80 |
| Basophils, $10^9$/l | 0.18 | 0.19 | 0.02 | 0.64 | 0.19 | 0.17 | 0.02 | 0.05 | 0.21 | 0.14 | 0.02 | < 0.01 |
| Basophils, % | 2.24 | 2.12 | 0.15 | 0.73 | 2.19 | 2.18 | 0.15 | 0.99 | 2.64 | 1.73 | 0.18 | 0.26 |
| IgG[4], titer | 4.54 | 4.39 | 0.18 | 0.56 | 4.36 | 4.58 | 0.18 | 0.38 | 4.60 | 4.34 | 0.22 | 0.47 |
| IgM[5], titer | 5.31 | 5.18 | 0.21 | 0.37 | 5.16 | 5.33 | 0.21 | 0.62 | 5.12 | 5.38 | 0.21 | 0.59 |
| Haptoglobin, mg/ml | 0.34 | 0.44 | 0.04 | 0.20 | 0.44 | 0.34 | 0.04 | 0.07 | 0.34 | 0.44 | 0.04 | 0.26 |
| Bilirubin, µmol/l | 12.77 | 10.37 | 0.62 | 0.01 | 12.71 | 10.43 | 0.87 | 0.16 | 10.82 | 12.32 | 0.62 | 0.05 |
| Cortisol, ng/ml | 10.79 | 8.79 | 0.99 | 0.03 | 10.35 | 9.23 | 0.99 | 0.34 | 7.95 | 11.63 | 0.96 | < 0.01 |

[1]SEM = standard error of the mean

[2]WBC = white blood cell count

[3]% = all proportions are relative to WBC

[4]IgG = immunoglobulin G

[5]IgM = immunoglobulin M.

**Table 5. Effects of diet composition (electrolytes vs milk) at the collection center, transport conditions (conditioned vs open truck) and transport duration (6 vs 18 hours) on different cell subsets of young calves analyzed with flow cytometry measured directly post-transport (T0) (LS means).**

| Parameter (%[1]) | Pre-transport diet | | | | Transport condition | | | | Transport duration | | | |
|---|---|---|---|---|---|---|---|---|---|---|---|---|
| | Electrolytes | Milk | SEM[2] | *P*-value | Conditioned | Open | SEM | *P*-Value | 6 h | 18 h | SEM | *P*-Value |
| CD8+ T cells | 8.45 | 7.26 | 0.41 | 0.05 | 7.63 | 8.10 | 0.41 | 0.35 | 6.90 | 8.85 | 0.50 | 0.01 |
| CD8+ perf+[3] | 1.10 | 1.24 | 0.11 | 0.66 | 1.17 | 1.17 | 0.11 | 0.99 | 0.96 | 1.38 | 0.14 | 0.36 |
| NK cells | 7.99 | 7.39 | 0.54 | 0.33 | 7.75 | 7.63 | 0.54 | 0.84 | 8.72 | 6.65 | 0.65 | 0.07 |
| NK perf+ | 1.66 | 1.65 | 0.07 | 0.98 | 1.70 | 1.61 | 0.07 | 0.76 | 1.65 | 1.66 | 0.09 | 0.95 |
| CD4+ T cells | 19.24 | 18.59 | 0.93 | 0.58 | 18.55 | 19.28 | 0.93 | 0.56 | 17.55 | 20.28 | 1.14 | 0.13 |
| CD4+ perf+ | 0.17 | 0.18 | 0.02 | 0.95 | 0.20 | 0.15 | 0.02 | 0.61 | 0.17 | 0.18 | 0.03 | 0.86 |
| δγ+ T cells | 39.89 | 42.24 | 1.69 | 0.31 | 41.30 | 40.83 | 1.69 | 0.77 | 42.93 | 39.20 | 2.07 | 0.25 |
| δγ+ perf+ | 0.17 | 0.21 | 0.02 | 0.70 | 0.21 | 0.17 | 0.02 | 0.69 | 0.18 | 0.20 | 0.02 | 0.87 |
| Monocytes | 40.99 | 45.09 | 1.80 | 0.11 | 44.86 | 41.22 | 1.80 | 0.16 | 41.62 | 44.47 | 2.20 | 0.44 |
| B cells | 2.28 | 2.69 | 0.21 | 0.16 | 2.50 | 2.47 | 0.21 | 0.90 | 2.84 | 2.13 | 0.25 | 0.08 |

[1]% = proportion relative to lymphocytes

[2]SEM = standard error of the means

[3]Perf+ = CD8+ T cells, NK cells, CD4+ and δγ+ T cells were stimulated also with perforin to examine the functionality of these cells and how their functionality was affected by the different treatments.

**Table 6. Effects of diet composition (electrolytes vs milk) at the collection center (CC), transport condition (conditioned vs open truck) and transport duration (6 vs 18 hours) on the difference between CC and post-transport (T0) immunological measurements (deltas, Δ = T0 −CC) of young veal calves (LS means).**

| Parameter | Pre-transport diet | | | | Transport condition | | | | Transport duration | | | |
|---|---|---|---|---|---|---|---|---|---|---|---|---|
| | Electrolytes | Milk | SEM[1] | *P*-value | Conditioned | Open | SEM | *P*-Value | 6 h | 18 h | SEM | *P*-Value |
| WBC[2], 10⁹/l | -1.90 | -1.35 | 0.51 | 0.39 | -0.98 | -2.27 | 0.51 | 0.04 | -1.91 | -1.34 | 0.60 | 0.53 |
| Monocytes, 10⁹/l | -0.26 | -0.19 | 0.09 | 0.42 | -0.11 | -0.34 | 0.09 | < 0.01 | -0.26 | -0.19 | 0.10 | 0.55 |
| Monocytes, %[3] | 0.04 | -0.11 | 0.42 | 0.95 | -0.05 | -0.03 | 0.42 | 0.98 | -0.27 | 0.20 | 0.42 | 0.99 |
| Lymphocytes, 10⁹/l | -0.40 | -0.31 | 0.11 | 0.51 | -0.29 | -0.43 | 0.11 | 0.32 | -0.40 | -0.31 | 0.13 | 0.65 |
| Lymphocytes, % | 2.72 | 1.49 | 1.14 | 0.92 | -0.22 | 4.45 | 1.11 | 0.81 | 1.81 | 2.39 | 1.08 | 0.99 |
| Neutrophils, 10⁹/l | -1.39 | -1.01 | 0.37 | 0.44 | -0.72 | -1.68 | 0.37 | 0.05 | -1.46 | -0.93 | 0.44 | 0.45 |
| Neutrophils, % | -6.07 | -3.46 | 1.28 | 0.90 | -2.63 | -6.91 | 1.27 | 0.76 | -4.25 | -5.26 | 1.27 | 0.95 |
| Eosinophils, 10⁹/l | 0.16 | 0.21 | 0.09 | 0.69 | 0.25 | 0.12 | 0.09 | 0.34 | 0.19 | 0.17 | 0.11 | 0.92 |
| Eosinophils, % | 2.33 | 2.14 | 0.54 | 0.87 | 2.21 | 2.25 | 0.54 | 0.97 | 2.77 | 1.68 | 0.54 | 0.71 |
| Basophils, 10⁹/l | -0.06 | -0.05 | 0.02 | 0.62 | -0.07 | -0.03 | 0.02 | 0.55 | -0.05 | -0.06 | 0.02 | 0.70 |
| Basophils, % | -0.34 | -0.05 | 0.22 | 0.89 | -0.61 | 0.23 | 0.22 | 0.57 | -0.06 | -0.32 | 0.22 | 0.90 |
| IgG[4], titer | -0.15 | -0.30 | 0.05 | 0.05 | -0.25 | -0.20 | 0.05 | 0.56 | -0.21 | -0.24 | 0.06 | 0.76 |
| IgM[5], titer | 0.10 | 0.18 | 0.31 | 0.86 | 0.09 | 0.20 | 0.31 | 0.81 | 0.48 | -0.19 | 0.39 | 0.29 |
| Haptoglobin, mg/ml | -0.01 | 0.03 | 0.03 | 0.26 | 0.05 | -0.03 | 0.03 | 0.06 | -0.01 | 0.03 | 0.04 | 0.50 |
| Bilirubin, μmol/l | 2.57 | -0.96 | 0.78 | < 0.01 | 1.19 | 0.42 | 0.78 | 0.49 | -0.06 | 1.66 | 0.96 | 0.28 |
| Cortisol, ng/ml | 0.48 | -2.01 | 1.13 | 0.07 | -1.29 | -0.23 | 1.13 | 0.43 | -3.62 | 2.09 | 1.32 | < 0.01 |

[1]SEM = standard error of the means

[2]WBC = white blood cell count

[3]% = all proportions are relative to WBC

[4]IgG = immunoglobulin G

[5]IgM = immunoglobulin M.

monocytes ($\Delta = 0.21 \times 10^9$/l), eosinophils ($\Delta = 0.24 \times 10^9$/l) and basophils ($\Delta = 0.02 \times 10^9$/l) than calves in the open truck. Despite the higher WBC, the effects of transport conditions on proportions of cells were different compared to those on absolute numbers. In calves transported in the conditioned truck, the higher WBC contributed to an increased proportion of neutrophils ($\Delta = 3.84\%$) but to a lower proportion of lymphocytes ($\Delta = -4.73\%$) in the blood compared to calves transported in the open truck. Transport condition had no significant effects on proportions of NK cells, B cells, and different T cells subsets upon arrival at T0.

Transport duration significantly affected basophils and cortisol concentrations (Table 4). Long-term transport (18 hours) led to lower basophils and higher cortisol values compared to 6 hours transport ($\Delta = -0.07 \times 10^9$/l and $\Delta = 3.68$ ng/ml, respectively). In addition, at T0, calves transported for 18 hours had a greater proportion of CD8+ lymphocytes in the blood than calves with 6 hours transport ($\Delta = 1.95\%$) (Table 5).

No differences were observed in perforin expression in NK cells, γδ T cells, CD8+, CD4+ T cells, indicating that the function of those cell subsets in the blood was not influenced by the different treatments (Table 5).

**Effects on the difference between pre- and post-transport measurements ($\Delta = T0 – CC$).** Pre-transport diet had an effect on differences in IgG titer and bilirubin concentrations between pre and post-transport (Table 6). IgG titer decreased during transport to a greater extent in milk-fed calves than in electrolyte-fed calves. Transport condition influenced the absolute amount of WBC, neutrophils and monocytes. Calves in the conditioned truck had a smaller decrease in WBC, neutrophils and monocytes than calves in the open truck (Table 6). Transport duration had no effects on deltas of differential white blood cell count.

Significant effects of pre-transport diet, transport duration or transport condition on deltas of NK cells, B cells, different T cells subsets were not present, nor on perforin expression (S1 Table).

## Effects of treatments in time until week 5 post-transport

**Interaction between pre-transport diet and time.** An interaction between pre-transport diet and time was found for bilirubin (Fig 1). Electrolyte-fed calves showed higher bilirubin concentrations at T0 than milk-fed calves, whereas at the other moments no differences were found.

**Interaction between transport condition and time.** Haptoglobin showed an interaction between transport condition and time (Fig 2). At T0 ($P < 0.05$) and at T24 ($P < 0.10$) calves transported in the conditioned truck showed higher haptoglobin levels than calves transported in the open truck, whereas at the other moments no significant differences were found between transport conditions.

**Interaction between transport duration and time.** An interaction between transport duration and time was detected for bilirubin and cortisol (Fig 3A and 3B). Calves transported for 18 hours had higher bilirubin at T0 and higher cortisol at T0 and T24 than calves transported for 6 hours, whereas at the other moments no differences between transport durations were found.

## Discussion

### Effects of pre-transport diet

In our experiment, electrolyte-fed calves had a higher cortisol concentration at arrival at the veal farm than milk-fed calves, probably due to the different composition of the pre-transport diet. It has been shown that cortisol can be influenced by the digestive functions that occur after feeding and by intestinal motility [22]. Plasma glucose and insulin are negatively

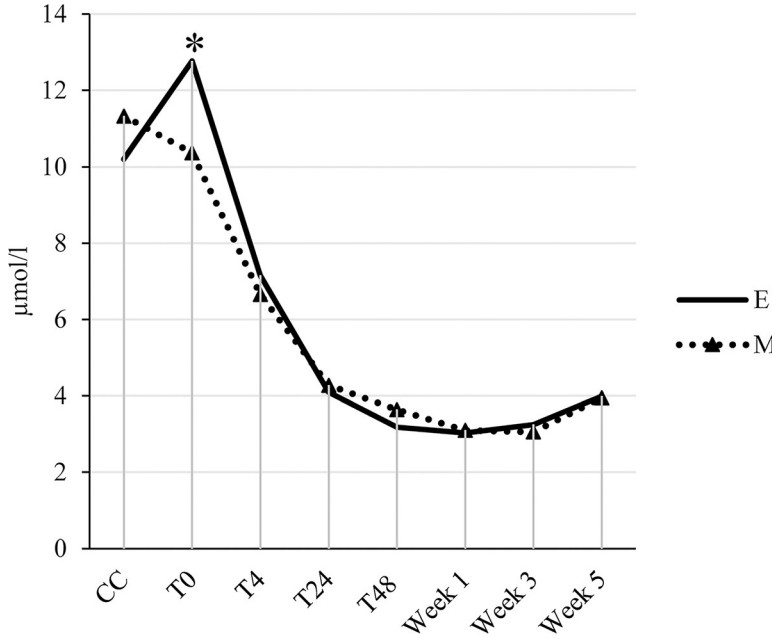

**Fig 1. Interaction between pre-transport diet (milk vs electrolytes) and time relative to transport for bilirubin in young veal calves.** Blood samples were collected at the collection center (CC), immediately post-transport (T0) and 4 (T4), 24 (T24), 48 (T48) hours, and week 1, 3, and 5 post-transport. Asterisks indicate significant differences ($P \leq 0.05$) between treatments within a sampling moment.

correlated to plasma ACTH and cortisol [23], and, hence, it is expected that glucose and insulin levels are higher and cortisol levels are lower in milk-fed animals in the current study. Moreover, feeding electrolytes might have contributed to a more negative energy balance

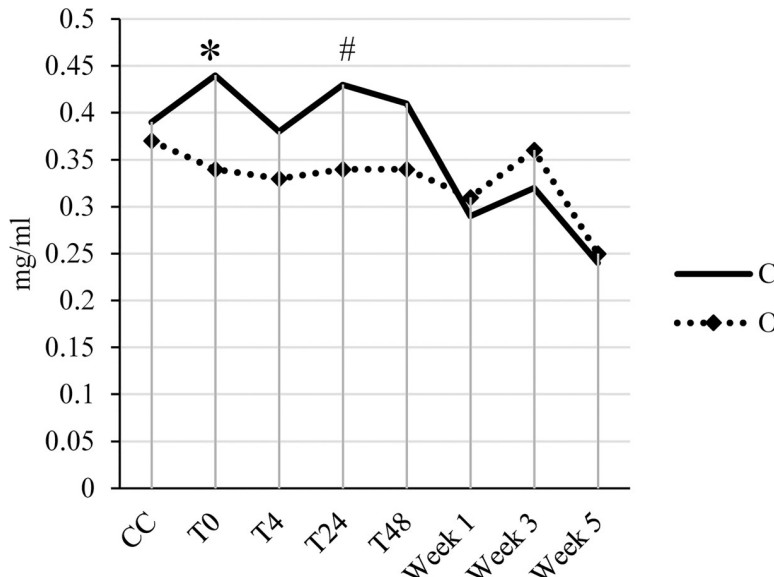

**Fig 2. Interaction between transport condition (conditioned truck vs open truck) and time relative to transport for haptoglobin in young veal calves.** Blood samples were collected at the collection center (CC), immediately post-transport (T0) and 4 (T4), 24 (T24), 48 (T48) hours, and week 1, 3, and 5 post-transport. Asterisks indicate significant differences ($P \leq 0.05$) between treatments within a sampling moment, whereas hashtags indicate a tendency towards significance ($0.05 \leq P \leq 0.10$).

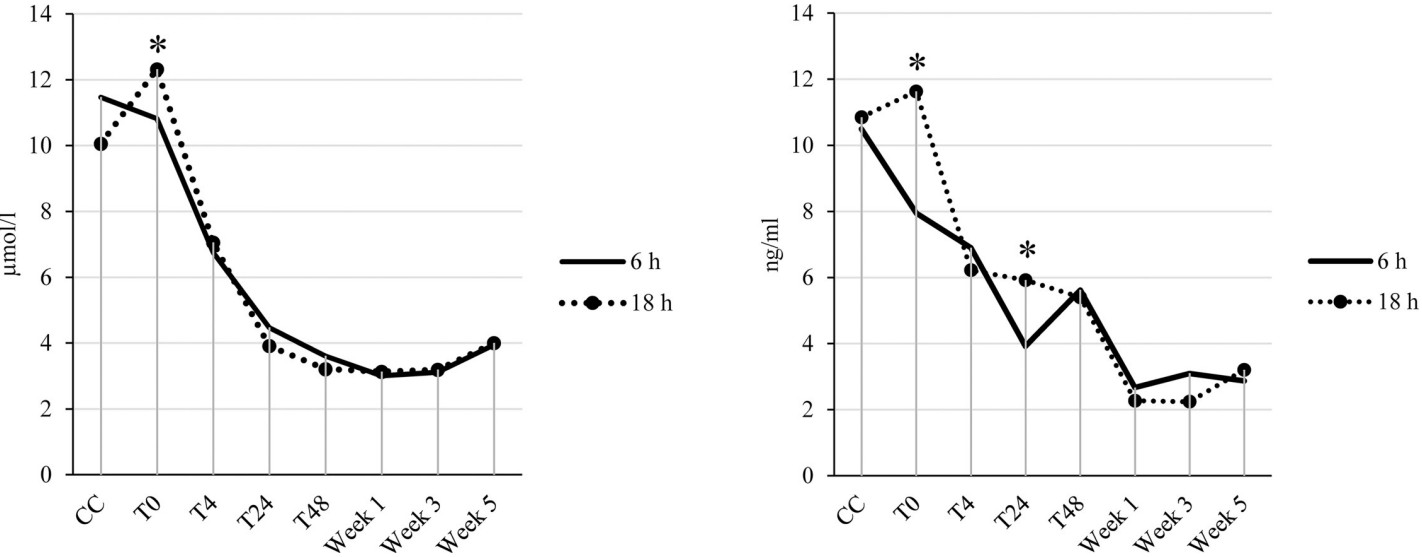

**Fig 3. Interactions between transport duration (6 hours vs 18 hours) and time relative to transport in young veal calves.** (A) Bilirubin and (B) Cortisol. Blood samples were collected at the collection center (CC), immediately post-transport (T0) and 4 (T4), 24 (T24), 48 (T48) hours, and week 1, 3, and 5 post-transport. Asterisks indicate significant differences ($P \leq 0.05$) between treatments within a sampling moment.

(NEB) and an enhanced feeling of hunger compared to feeding milk. Hunger increases corticotropin-releasing factor secretion and consequently cortisol secretion in calves [22].

In the current experiment, pre-transport diet also had an effect on absolute levels of monocytes and CD8+ lymphocytes at arrival at the veal farm. Nutrition has been shown to play a key role in immune responses of young calves [24,25]. Especially the energy and protein intake in calves during the first weeks after birth, can influence cell-mediated immunity, cytokine production, phagocytic function and secretory IgA antibody concentrations, but also specific absolute numbers of monocytes [26]. However, these studies were conducted in young calves at the dairy farms, without any challenge or transport, so probably their values were closer to baseline values than in the current study.

In addition, Murata et al. [27] and Earley et al. [4] reported increased counts of monocytes and differences in lymphocyte subsets post-transport. This effect was attributed to transport duration rather than to pre-transport diet. To our knowledge, there are no other studies conducted on effects of pre-transport diet on subsets of these immune cells of young calves on arrival at the veal farm. It can be hypothesized that the higher counts of monocytes of milk-fed calves may have a protective function against environmental microorganisms, thus reducing the risk of diseases in calves.

## Interaction between pre-transport diet and transport duration

**Effects on measurements immediately post-transport (T0).** Bilirubin is commonly used as a biomarker of liver status in cattle, especially in dairy cows [28,29]. In cows, higher bilirubin concentrations are associated with a lower clearance of secretory enzymes in the liver as a response to liver cell damage [29,30]. In addition, bilirubin is positively correlated with the degree of fat infiltration in the liver and high bilirubin levels are related with a more negative energy balance (NEB) [31]. In our study, the high bilirubin values after prolonged transport in both milk-fed and electrolyte-fed calves might be mainly caused by a prolonged fasting period. Both feeding strategies at the collection center might not have provided calves enough energy to cover the entire period of travel. Therefore, calves were likely in a NEB at T0. This is

supported by the electrolyte-fed calves after 6 hours transport, which also demonstrated high bilirubin concentrations, probably because of the low energy intake, whereas milk-fed calves showed lower bilirubin concentrations probably because they had more energy intake, covering the energy needs during a 6 hours transport duration.

**Effects on the difference between pre- and post-transport measurements (Δ = T0 – CC).** Basophils have numerous functions, including a beneficial role in protective immunity against parasitic infections and a role in autoimmune and inflammatory diseases [32]. Calves have generally a low basophil count in their blood [33]. An increase in basophils can be indicative of a stress response, following transport in adult cattle [4,27]. Transport related-stress appears to be associated with cell trafficking and redistribution of peripheral lymphocytes between different immune compartments [34]. Thus, it can be hypothesized that the increase in basophils was triggered by redistribution of lymphocytes in the peripheral blood. However, to our knowledge, there are no studies that investigated effects of transport on these cells in young calves. Additionally, there is no clear evidence that diet could be a factor that contributes to an increase in number of basophils. It could be hypothesized that the higher protein and energy content of milk compared to electrolytes may have modified the adhesion marker expression on T lymphocytes and may have resulted in an increase in basophil counts.

Bovine haptoglobin is an APP that exerts numerous biological functions. In cattle, the primary function of haptoglobin is to form complexes with free hemoglobin in the blood in order to prevent oxidative damages [35,36]. An increase in haptoglobin has been reported in calves with a higher disease incidence [37]. Moreover, a rise in haptoglobin concentrations was also found in calves after long distance transport [17,35]. Therefore, these studies suggested that haptoglobin can be used also as a marker of health status or stress in calves. In our study, calves transported for 18 hours and fed with milk or electrolytes showed a different response in terms of haptoglobin concentrations. The increase in haptoglobin after 18 hours transport in milk-fed calves might suggest that these animals were probably experiencing higher post-transport stress than the electrolyte-fed calves. Additionally, in our study, the increase in haptoglobin was relatively small compared to acute values reported in the literature (1.62 ± 0.47 g/l).

## Effects of transport duration

In our study, effects of transport duration were also present on absolute values and on deltas of cortisol between T0 and CC. The higher cortisol concentrations after 18 hours transport were in line with previous studies. Knowles et al. [38] found increased cortisol concentrations post-transport up to 2.3 μg/100 ml in cattle compared to pre-transport concentrations. Averos et al. [39] showed that cortisol increased when young bulls were transported for 13 hours from a collection center (6.5 ng/ml) to a growing-finishing farm (12.0 ng/ml). However, other studies on transport related effects on cortisol showed ambiguous results. Odore et al. [40] found already a significant increase in cortisol concentrations after short-term transport duration. Honkavaara et al. [41] also observed that animals transported for short journeys had a higher cortisol than animals transported for long journeys (up to 14 hours). Short transport duration can cause acute psychological stress due to the novelty of the transport process, whereas during long transport duration, the exhaustion of the adrenal gland may occur [40]. Kent and Ewbank [42], and Warriss et al. [43] suggested that the decrease in cortisol concentrations during long transport duration might also be due to the adaptation of cattle to the transport.

All these studies used older animals than those included in the current study, which makes a comparison between previous results and the present findings in young calves difficult. Young animals still have to complete their HPA axis development (see Marcato et al. [44] for overview), and consequently, the responses to stress in young animals might be different and

not really consistent as the ones of mature cattle [2,13]. Overall, in our study, it seems that calves had elevated levels of cortisol directly and 24 hours post-transport.

Following transport, an increase in the adrenal cortex activity might affect the immune system [40,45]. Some studies reported that transport can lead to lower number of immune cells in blood, whereas others suggested that transport can have stimulating effects on the immune system, especially after acute stress [10,45]. Lymphocytes express measurable concentrations of glucocorticoid and adrenergic receptors, which can be down-regulated or altered during a stress response [46–48]. Riondato et al. [10] showed a decrease in proportions of calves' CD4+ and CD8+ T cells in blood after 14 hours transport duration, whereas the decrease in CD21+ B cells occurred just a day later. These data are different from the results of the present study, which demonstrated no effects on CD4+ T cells, but higher proportions of CD8+ T cells after 18 hours transport duration compared to 6 hours transport duration. This difference might reflect the activation of the immune system to redistribute more T cells from the secondary lymphoid organ into the peripheral blood. Masmeijer et al. [49] indicated that, in young calves between 14 and 28 days of age, transport-related higher glucocorticoids may cause a robust leukocyte redistribution by inducing increased expression of surface receptors (e.g. CD172a, CD11a) on stress-activated monocytes. A higher humoral and cellular response might be related to a decrease in susceptibility of calves to infections [50]. In the current experiment, a tendency towards lower amount of NK cells and B cells after 18 h transport was also found compared to 6 hours transport. These data might be explained by higher concentrations of cortisol after long transport duration. However, Ishizaki and Kariya [51], found a positive correlation ($r = 0.704$; $P < 0.01$) between cortisol concentrations and NK cell counts in the blood. The study was conducted on short-term transport and acute stress effects, thus the effects might be different for long-term transport, which might have an inhibitory effect on these cells.

## Effects of transport conditions

In the current study, immediately post-transport, a higher number of WBC, a higher proportion of neutrophils, and a lower proportion of lymphocytes were found after transport in the conditioned truck compared to the open truck. Earley et al. [4] indicated that neutrophilia in conjunction with lymphopenia are common responses observed following transport in all age groups of cattle. Ishizaki and Kariya [51] reported that higher glucocorticoids levels result in lymphocyte destruction in the thymus cortex and in the extension of the neutrophil half-life. These changes might be indicators of transport stress, but due to a lack of literature on transport conditions, potential reasons for the differences between the open and the conditioned truck remain unknown. In the presence of a pathogen, a higher number of neutrophils might also be the cause of the excessive inflammation and tissues damage found in cattle with bovine respiratory disease following transport [52,53]. Transport stress might also lead to other shifts in immune cell subsets, including an increase in monocytes, eosinophils and basophils [27,54]. In fact, transport of young calves (14 until 28 days of age) is associated with an increased production in glucocorticoids, which can cause a robust leukocyte redistribution [49]. This general mechanism of redistribution may also explain the current effects of the conditioned truck on total WBC and some of the related cells. Perhaps, a range of environmental factors may be involved, such as draught, temperature, relative humidity, or vibrations inside the vehicle. However, this needs to be investigated further.

## Effects of factors on variables until week 5 post-transport

In our study, the recovery rate of calves after 6 hours transport was similar to the one after 18 hours transport, since calves restored their pre-transport values within 4 hours post-transport.

However, calves transported for 6 hours had a decrease in cortisol concentrations immediately post-transport, suggesting that short transport duration has no effect on the HPA axis response. This result is in agreement with the studies of Cole et al. [55] and Mitchell et al. [56] who found lower cortisol concentration post-transport than pre-transport. Moreover, at T24, concentrations of cortisol in calves transported for 6 hours were slightly lower than those of calves transported for 18 hours, suggesting that the recovery rate of calves with long transport duration took a longer time period.

Bilirubin concentrations after 6 and 18 hours transport followed the same trend as cortisol, thus the recovery rate of calves, based on this parameter, seemed also fast. It seems that at 48 hours post-transport all values were similar regardless of treatment, thus these values might represent baseline values. However, since calves were already challenged before transport, it remains unknown whether these values are representative of reference values for homeostasis.

Immediately post-transport, haptoglobin increased in calves transported in the conditioned truck, whereas it decreased in calves transported in the open truck. The same concentrations were also visible 24 hours post-transport, where haptoglobin concentrations were gradually lower until week 5 post-transport. Higher haptoglobin concentrations are associated with tissue injury, inflammation or infection [16,57]. Other studies showed an increase in haptoglobin in response to stress of transport [15,16]. Thus, the increase in haptoglobin found in calves transported in the conditioned truck might be related with a greater transport stress experienced by calves. The recovery rate of calves was slower compared to the one for cortisol and bilirubin, because haptoglobin tended to be high 24 hours post-transport. However, it is difficult to make a comparison with the recovery rate of calves used in other studies due to a lack of reference values for different time points after transport. Overall, it can be concluded that there are no long-term (> 48 hours post-transport) effects of treatments on bilirubin, cortisol and haptoglobin concentrations.

## Conclusions

The current study showed that all treatments (pre-transport diet, transport duration and transport condition) affected different parts of the immune system of young veal calves. The higher energy content of milk relative to electrolytes likely prevented the rise in bilirubin concentrations in calves transported for 6 hours. Long transport duration compared to short transport duration contributed to higher blood cortisol concentrations in calves post-transport. We hypothesized that the higher stress and cortisol concentrations in calves transported for 18 hours may have resulted in a redistribution of leukocytes in their circulation, leading to higher CD8+ T cells than in calves transported for 6 hours. At present it remains unclear which transport factors (e.g. temperature, humidity, draught) are most important in relation to potential effects on the immune system of young veal calves. The potential implications of differences in immune cells upon arrival at the veal farm for later clinical health and disease incidence during the rearing period remains to be determined, and will be the subject of follow-up research.

## Supporting information

**S1 Fig.** Design of the truck and the trailer for batch 1 (A) and batch 2 (B) of calves. M6C = milk, 6 hours transport, conditioned truck; M18C = milk, 18 hours transport, conditioned truck; M6O = milk, 6 hours transport, open truck; M18O = milk, 18 hours transport, open truck; E6C = electrolytes, 6 hours transport, conditioned truck; E18C = electrolytes, 18 hours transport, conditioned truck; E6O = electrolytes, 6 hours transport, open truck; E18O = electrolytes, 18 hours transport, open truck. (PDF)

**S1 Table. Effects of diet composition (electrolytes vs milk) at the collection center (CC), transport condition (conditioned vs open truck) and transport duration (6 vs 18 hours) on the difference between CC and post-transport (T0) measurements (deltas, Δ = T0 –CC) of different cell subsets in blood of young veal calves (LS means).** [1]% = proportion relative to lymphocytes; [2]SEM = standard error of the means; [3]Perf+ = CD8+ T cells, NK cells, CD4+ and δγ+ T cells were stimulated with perforin to examine the functionality of these cells and how their functionality was affected by the different treatments.
(PDF)

**S1 Data.**
(XLSX)

## Acknowledgments

The authors would like to acknowledge representatives of the Dutch veal and livestock transport industries for providing personal support and facilities, and Rimondia B.V. for help during blood sampling and analyses. Furthermore, the authors would like to thank Ger de Vries-Reiligh and Rudie Koopmanschap, Bjorge Lauressen, Monique Ooms, Ilona van den Anker, Manon van Marwijk and Joop Aarts, for their help with cell isolation and with the analyses. We also want to thank Henk Gunnink, Theo van Hattum, Henny Reimert and Joop van der Werf for their skilled assistance during the experiment.

## Author Contributions

**Conceptualization:** Francesca Marcato.

**Data curation:** Francesca Marcato, Maaike Wolthuis-Fillerup.

**Formal analysis:** Francesca Marcato.

**Funding acquisition:** Kees van Reenen.

**Investigation:** Francesca Marcato, Christine A. Jansen, Maaike Wolthuis-Fillerup.

**Methodology:** Francesca Marcato, Bas Engel, Kees van Reenen.

**Project administration:** Kees van Reenen.

**Resources:** Henry van den Brand, Kees van Reenen.

**Supervision:** Henry van den Brand, Kees van Reenen.

**Validation:** Henry van den Brand, Kees van Reenen.

**Visualization:** Francesca Marcato, Henry van den Brand, Victor P. M. G. Rutten, Kees van Reenen.

**Writing – original draft:** Francesca Marcato.

**Writing – review & editing:** Francesca Marcato, Henry van den Brand, Christine A. Jansen, Victor P. M. G. Rutten, Bas Kemp, Bas Engel, Maaike Wolthuis-Fillerup, Kees van Reenen.

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
