## [Decision Letter · Decision Letter 0]

7 Dec 2020

PONE-D-20-32019

Effects of pre-transport diet, transport duration and transport condition on immune cell subsets, haptoglobin, cortisol and bilirubin in young veal calves

PLOS ONE

Dear Dr. Marcato,

Thank you for submitting your manuscript to PLOS ONE. After careful consideration, we feel that it has merit but does not fully meet PLOS ONE’s publication criteria as it currently stands. Therefore, we invite you to submit a revised version of the manuscript that addresses the points raised during the review process.

We look forward to receiving your revised manuscript.

Kind regards,

Juan J Loor

Academic Editor

PLOS ONE

Journal Requirements:

2. In your Methods section, please state the volume of the blood samples collected for use in your study.

3. In your Methods section, please include a comment about the state of the animals following this research. Were they euthanized or housed for use in further research? If any animals were sacrificed by the authors, please include the method of euthanasia and describe any efforts that were undertaken to reduce animal suffering.

Reviewers' comments:

Reviewer's Responses to Questions

**Comments to the Author**

1. Is the manuscript technically sound, and do the data support the conclusions?

Reviewer #1: Yes

2. Has the statistical analysis been performed appropriately and rigorously? 

Reviewer #1: Yes

3. Have the authors made all data underlying the findings in their manuscript fully available?

Reviewer #1: Yes

4. Is the manuscript presented in an intelligible fashion and written in standard English?

Reviewer #1: No

5. Review Comments to the Author

Reviewer #1: The aim of this study was to investigate effects of pre-transport diets, transport durations and transport conditions on immune cell subsets, haptoglobin, cortisol and bilirubin of young calves upon arrival at the veal farm. This reviewer considers that the manuscript is suitable for publication. Materials and methods were extensively described, specially the statistical analysis and results were properly discussed. I only have some grammar corrections as follows:

Line 28: Please, separate the word fowcytometry

Lines 85 – 89: please, compile these two sentences in one sentence.

Line 146, 235, 268, 401, 412, 449, 486 and 591: please, change centre for center.

Line 162: Americans spell it "titer" while the English spell it "titre".

Line 194: Please change “permeabilized with a permeabilization” for “permeabilised with a permeabilisation”.

Line 199: Same as above

Table 4, 5 and 6 title: please, change centre for center.

Line 463: Please, change “However, there are to our knowledge no studies” to “However, to our knowledge, there are no studies…”

Line 576: Please, change hypothesized for hypothesized

Definition and clarity of main images must be improved.

6. PLOS authors have the option to publish the peer review history of their article (what does this mean?). If published, this will include your full peer review and any attached files.

Reviewer #1: No

---

## [Author Response · Author response to Decision Letter 0]

22 Jan 2021

Dear Sir/Madam,

Thank you for reserving time to judge the paper and for your feedback. Please find below the list of changes we have made in the manuscript according to your comments. Our answers are indicated as Authors. These answers can also be found in the attached document "Response to Reviewers".

Journal requirements

Authors: We have checked the manuscript and it meets PLOS ONE's style requirements. We have made few corrections in L 277- 279 in the new manuscript.

2. In your Methods section, please state the volume of the blood samples collected for use in your study.

Authors: We have added the volume of the blood samples collected during the experiment (see L148 in the new manuscript).

3. In your Methods section, please include a comment about the state of the animals following this research. Were they euthanized or housed for use in further research? If any animals were sacrificed by the authors, please include the method of euthanasia and describe any efforts that were undertaken to reduce animal suffering.

Authors: The animals included in the current research followed the normal production cycle of the veal industry, at the end of which calves were slaughtered and their meat was used for human consumption (see L136-138 in the new manuscript).

Reviewer 1

1. Is the manuscript technically sound, and do the data support the conclusions?

Reviewer #1: Yes

2. Has the statistical analysis been performed appropriately and rigorously? 

Reviewer #1: Yes

3. Have the authors made all data underlying the findings in their manuscript fully available?

Reviewer #1: Yes

Authors: We would like to provide a new version of the Data Availability Statement (please check the cover letter).

4. Is the manuscript presented in an intelligible fashion and written in standard English?

 Reviewer #1: No

Authors: We have checked the manuscript and made all corrections raised by the reviewer.

5. Review Comments to the Author

Reviewer #1: The aim of this study was to investigate effects of pre-transport diets, transport durations and transport conditions on immune cell subsets, haptoglobin, cortisol and bilirubin of young calves upon arrival at the veal farm. This reviewer considers that the manuscript is suitable for publication. Materials and methods were extensively described, specially the statistical analysis and results were properly discussed. I only have some grammar corrections as follows:

Line 28: Please, separate the word fowcytometry 

Authors: we have corrected the word in the manuscript (see L28 in the new manuscript).

Lines 85 – 89: please, compile these two sentences in one sentence.

Authors: we have combined the two sentences into one (see L 85-88 in the new manuscript).

Line 146, 235, 268, 401, 412, 449, 486 and 591: please, change centre for center.

Authors: we have corrected the word in the manuscript (see L146, 235, 268, 401, 412, 449, 486 and 591 in the new manuscript).

Line 162: Americans spell it "titer" while the English spell it "titre".

Authors: we have corrected the word in the manuscript (see L162 in the new manuscript).

Line 194: Please change “permeabilized with a permeabilization” for “permeabilised with a permeabilisation”.

Authors: we have corrected the words in the manuscript (see L194 in the new manuscript).

Line 199: Same as above

Authors: we have corrected the word in the manuscript (see L199 in the new manuscript).

Table 4, 5 and 6 title: please, change centre for center.

Authors: we have corrected the word in tables 4, 5, 6 (see L347, 351 and 380 in the new manuscript).

Line 463: Please, change “However, there are to our knowledge no studies” to “However, to our knowledge, there are no studies…”

Authors: we have corrected the sentence (see L463-464 in the new manuscript).

Line 576: Please, change hypothesized for hypothesized

Authors: we have corrected the word (see L576 in the new manuscript).

Definition and clarity of main images must be improved.

Authors: we have improved the quality, definition and clarity of the figures (please see new attached files).

---

## [Editor Report · Decision Letter 1]

29 Jan 2021

Effects of pre-transport diet, transport duration and transport condition on immune cell subsets, haptoglobin, cortisol and bilirubin in young veal calves

PONE-D-20-32019R1

Dear Dr. Marcato,

We’re pleased to inform you that your manuscript has been judged scientifically suitable for publication and will be formally accepted for publication once it meets all outstanding technical requirements.

Kind regards,

Juan J Loor

Academic Editor

PLOS ONE
---

## [Editor Report · Acceptance letter]

2 Feb 2021

PONE-D-20-32019R1 

Effects of pre-transport diet, transport duration and transport condition on immune cell subsets, haptoglobin, cortisol and bilirubin in young veal calves 

Dear Dr. Marcato:

I'm pleased to inform you that your manuscript has been deemed suitable for publication in PLOS ONE. Congratulations! Your manuscript is now with our production department. 

Kind regards, 

on behalf of

Dr. Juan J Loor 

Academic Editor

PLOS ONE